# Light-independent regulation of algal photoprotection by $CO_2$ availability

M. Águila Ruiz-Sola [1,8,9], Serena Flori [1,9], Yizhong Yuan [1,9], Gaelle Villain[1], Emanuel Sanz-Luque [2,3], Petra Redekop [2], Ryutaro Tokutsu [4], Anika Küken [5,6], Angeliki Tsichla[1], Georgios Kepesidis [1], Guillaume Allorent[1], Marius Arend [5,6], Fabrizio Iacono [1], Giovanni Finazzi [1], Michael Hippler [7], Zoran Nikoloski[5,6], Jun Minagawa [4], Arthur R. Grossman [2] & Dimitris Petroutsos [1] ✉

Photosynthetic algae have evolved mechanisms to cope with suboptimal light and $CO_2$ conditions. When light energy exceeds $CO_2$ fixation capacity, *Chlamydomonas reinhardtii* activates photoprotection, mediated by LHCSR1/3 and PSBS, and the $CO_2$ Concentrating Mechanism (CCM). How light and $CO_2$ signals converge to regulate these processes remains unclear. Here, we show that excess light activates photoprotection- and CCM-related genes by altering intracellular $CO_2$ concentrations and that depletion of $CO_2$ drives these responses, even in total darkness. High $CO_2$ levels, derived from respiration or impaired photosynthetic fixation, repress *LHCSR3*/CCM genes while stabilizing the LHCSR1 protein. Finally, we show that the CCM regulator CIA5 also regulates photoprotection, controlling *LHCSR3* and *PSBS* transcript accumulation while inhibiting LHCSR1 protein accumulation. This work has allowed us to dissect the effect of $CO_2$ and light on CCM and photoprotection, demonstrating that light often indirectly affects these processes by impacting intracellular $CO_2$ levels.

A major challenge for photosynthetic organisms is to efficiently acclimate to highly dynamic light and nutrient conditions that occur in natural environments. While light provides the energy that fuels photosynthetic $CO_2$ fixation, excess light can cause oxidative damage and ultimately result in cell death. Therefore, light absorption must be precisely managed via photoprotective mechanisms that help integrate the use of light energy with $CO_2$ availability and the potential of the organism to grow and store fixed carbon. A dominant photoprotective mechanism, called qE (energy-dependent quenching), results in the harmless dissipation of excess absorbed light energy as

heat[1,2]. Triggering qE requires the synthesis of specific proteins and pigments that are controlled both transcriptionally and posttranscriptionally.

In the green microalga *Chlamydomonas reinhardtii* (hereafter *Chlamydomonas*), qE depends on the nucleus-encoded, chloroplast-localized Light Harvesting Complex-Stress Related (LHCSR) proteins LHCSR1, LHCSR3 and Photosystem II Subunit S, PSBS, which are present in many algae and lower plants[3] and belong to the Light Harvesting Complex protein superfamily[4]. The *LHCSR3.1* and *LHCSR3.2* genes in *Chlamydomonas* encode identical LHCSR3 proteins[5], while

[1]Univ. Grenoble Alpes, CNRS, CEA, INRAE, IRIG-LPCV, 38000 Grenoble, France. [2]The Carnegie Institution for Science, Department of Plant Biology, Stanford, CA 94305, USA. [3]University of Cordoba, Department of Biochemistry and Molecular Biology, Cordoba, Spain. [4]Division of Environmental photobiology, National Institute for Basic Biology (NIBB), Nishigonaka 38, Myodaiji, Okazaki 444-8585, Japan. [5]Bioinformatics Group, Institute of Biochemistry and Biology, University of Potsdam, Potsdam, Germany. [6]Max-Planck-Institute of Molecular Plant Physiology, Potsdam, Golm, Germany. [7]Institute of Plant Biology and Biotechnology, Westfälische Wilhelms Universität, 48143 Münster, Germany. [8]Present address: Instituto de Bioquímica Vegetal y Fotosíntesis, Consejo Superior de Investigaciones Científicas-Universidad de Sevilla, Sevilla, Spain. [9]These authors contributed equally: M. Águila Ruiz-Sola, Serena Flori, Yizhong Yuan. ✉e-mail: dimitrios.petroutsos@cnrs.fr

*PSBS1* and *PSBS2* encode proteins that differ by only one amino acid of the chloroplast transit peptide[6]. While LHCSR1 and LHCSR3 are present in algae but not in vascular plants, PSBS is present in both[4]. PSBS in *Chlamydomonas* is transiently expressed in cells exposed to high light (HL)[6,7] and accumulates in cells exposed to UV-B irradiation[8]. LHCSR3 is the main qE effector protein in HL[5], although LHCSR1 can significantly contribute to qE under certain conditions[9]. In *Chlamydomonas*, expression of *LHCSR3* has been reported to increase upon absorption of blue-light by the photoreceptor phototropin (PHOT1)[10] and involves calcium ion signaling[11], active photosynthetic electron transport (PET)[10,11] and the transcriptional factor CONSTANS, which is also required for activation of the *LHCSR1* and *PSBS* genes[12,13].

Similar to the dynamic light cue, the concentration of inorganic carbon ($HCO_3^-$, $CO_2$ and $CO_3^{2-}$, together designated Ci) in aquatic environments varies spatially and temporally; aquatic $CO_2$ levels can also fluctuate from extremely high (hyper-saturated) to extremely low[14]. Because low $CO_2$ levels limit photoautotrophic growth, microalgae have evolved a $CO_2$ Concentrating Mechanism (CCM) that elevates the level of $CO_2$ at the site of fixation by Ribulose-1,5-bisphosphate carboxylase/oxygenase (RuBisCO). Major components of the CCM are carbonic anhydrases (CAH), which facilitate interconversions among the different Ci species, and Ci transporters. The genes encoding many Ci transporters and CAHs are under the control of the zinc-finger type potential transcription regulator CIA5 (also CCM1)[15,16], which is localized in the nucleus[17] and controls expression of low-$CO_2$ responsive genes.

In addition to the use of $CO_2$ to support phototrophic growth in the light, *Chlamydomonas* can also use the two-carbon molecule acetate either in the dark to support heterotrophic growth, or in the light, to support photoheterotrophic or mixotrophic growth[18]. Acetate is incorporated into acetyl-CoA either in a one-step reaction catalyzed by acetyl-CoA synthetase (ACS), or in two steps that use acetate kinase (ACK) and phosphate acetyltransferase (PAT), which sequentially catalyze the formation of acetyl-phosphate and acetyl-CoA[19]. Acetyl-CoA can then enter the glyoxylate cycle, a shunt of the tricarboxylic acid (TCA) cycle[20], recently characterized in *Chlamydomonas*[21], where it can be converted to metabolites that are used for anabolic metabolism. Alternatively, acetyl-CoA enters the TCA cycle to feed the respiratory chain with reducing equivalents. Both, the glyoxylate cycle and respiration are essential for growth in the dark since *Chlamydomonas* mutants affected in either of these processes are unable to grow heterotrophically[21,22].

Despite the evident connection between light and $CO_2$ levels, the physiological responses to different light and $CO_2$ availabilities have been traditionally studied separately. However, several lines of evidence indicate that both acetate and Ci abundance impact not only qE but also the establishment of the CCM in *Chlamydomonas*[23–26], while *LHCSR3* transcripts accumulation have been reported to be CIA5-dependent[26–28]. Yet, the mechanism(s) associated with carbon-dependent regulation of qE and CCM induction and the intimate link between the two processes have still not been clearly defined.

Here, using genetic and mathematical modelling approaches, we demonstrate that inhibition of LHCSR3 accumulation and CCM activity by acetate is at the level of transcription and a consequence of metabolically produced $CO_2$. We also show that exposure of *Chlamydomonas* to HL triggers not only HL responses, but also low-$CO_2$ responses, and we report the discovery of a novel $CO_2$- and CIA5-dependent mechanism that activates *LHCSR3* gene expression even in complete darkness. Finally, we propose that PET is critical for the activation of *LHCSR3* transcription because it sustains $CO_2$ fixation, consuming intracellular $CO_2$ and thereby relieving its inhibitory effect. This work emphasizes the importance of $CO_2$ in regulating photoprotection and the CCM, and demonstrates that light often indirectly affects these processes by altering intracellular $CO_2$ levels.

## Results

### $CO_2$ generated from acetate metabolism inhibits *LHCSR3*

To gain insights into the effect of carbon metabolism on photoprotection, we explored the impact of acetate and high $CO_2$ on *LHCSR3* mRNA and protein levels in wild-type (WT) cells and in two mutants impaired in acetate metabolism; the *icl* mutant, which lacks isocitrate lyase, a key enzyme of the glyoxylate cycle[21], and the *dum11* mutant, which is defective in the ubiquinol cytochrome c oxidoreductase (respiratory complex III)[29]. The presence of acetate in the medium of WT cells inhibited the accumulation of the *LHCSR3* transcript (Fig. 1a, note the logarithmic scale) in both low light (LL) and high light (HL) conditions. No protein was detected in WT under any condition in LL, but inhibition by acetate was apparent in HL (Fig. 1b), as previously reported[25]. However, in the *icl* mutant, acetate had no inhibitory effect on the accumulation of LHCSR3 mRNA (Fig. 1a) or protein (Fig. 1b) in either HL or LL, while the *icl::ICL*-complemented line, designated *icl-C*, exhibited similar behavior to that of WT cells (Fig. 1a, b). Additionally, acetate did not alter LHCSR3 transcript or protein accumulation in HL-treated *dum11* mutant cells (Fig. 1c, d), while under LL, acetate inhibited *LHCSR3* transcript in the *dum11* mutant but to a much smaller extent than in WT (Fig. 1c). Together, these results suggest that the acetate administered to the cells must be metabolized for it to have a suppressive effect on the accumulation of LHCSR3 transcript and protein in HL. We also sparged WT, *icl, icl-C* and *dum11* cells with 5% $CO_2$ both in LL and HL. $CO_2$ strongly repressed the accumulation of LHCSR3 mRNA and protein in all genotypes, including the metabolic mutants *icl* and *dum11* for which expression of *LHCSR3* was unaffected by acetate (Fig. 1a-d).

We evaluated the impact of carbon availability on the photosynthetic properties of cells. The presence of acetate in the medium of WT cells enhanced photosynthetic electron transport (rETR) and strongly suppressed qE (Supplementary Fig. 1). In the *icl* mutant, acetate enhanced the extent of rETR only by ~10% compared with 60% for WT cells. Additionally, acetate caused less pronounced suppression of qE in the *icl* mutant (by 40%) compared to the level of suppression in WT cells (by 95%); *icl-C*, behaved similarly to WT cells. As expected, $CO_2$ enhanced rETR and suppressed qE in WT, *icl* and *icl-C* (Supplementary Fig. 1).

The similarity between the impact of acetate and 5% $CO_2$ on *LHCSR3* expression in WT and *icl-C* cells (Fig. 1a) as well as on their photosynthetic properties (Supplementary Fig. 1) raised the possibility that both treatments elicited a common mechanism of *LHCSR3* control, possibly reflecting a change in the $CO_2$ concentration within the cell or growth medium. This possibility is plausible based on the finding that acetate metabolism leads to the generation of $CO_2$[30]. To investigate whether the generation of $CO_2$ via acetate metabolism can explain the repression of LHCSR3 transcript and protein levels, we monitored the levels of transcripts from the *RHP1* gene in the mutant and WT cells; *RHP1* (aka *RH1*) encodes a $CO_2$ channel shown to be $CO_2$ responsive and to accumulate in cells grown in a high $CO_2$ atmosphere[31]. Acetate or 5% $CO_2$ were introduced to WT and *icl* mutant cells acclimated in LL and air and the levels of the *LHCSR3* and *RHP1* transcripts were assayed over a period of 8 h in LL (Fig. 2a, b). The *LHCSR3* transcript accumulation patterns observed agreed with the findings presented in Fig. 1a (LL panel). In WT cells, acetate and $CO_2$ caused a reduction in *LHCSR3* mRNA accumulation over the LL period relative to the control (no acetate, air). Additionally, in the *icl* mutant, acetate did not affect the accumulation of this transcript while $CO_2$ efficiently repressed the *LHCSR3* transcript level (Fig. 2a). Under these experimental conditions, acetate levels in the medium decreased in WT cultures but remained unchanged in cultures of the *icl* mutant (Fig. 2c). Lastly, *RHP1* expression increased in WT and *icl* mutant cells when the culture was sparged with $CO_2$, but only in the WT cells when the cultures were not sparged with $CO_2$ and only supplemented with

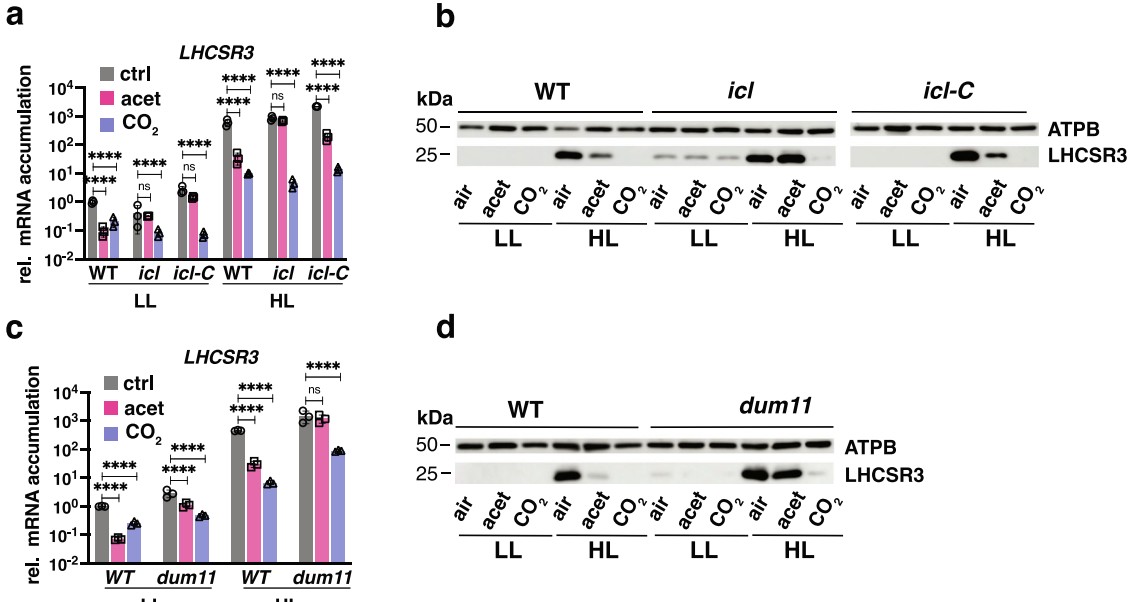

**Fig. 1 | Acetate needs to be metabolized to inhibit LHCSR3 accumulation.** WT, *icl*, *icl-C* and *dum11* strains were acclimated for 16 h in LL (15 μmol photons m⁻² s⁻¹) in HSM; sparged with air (labelled as "air"); sparged with air and supplemented with 10 mM sodium acetate (labelled as "acet"); sparged with air enriched with 5% $CO_2$ (labelled as "$CO_2$"). After sampling for the LL conditions, light intensity was increased to 600 μmol photons m⁻² s⁻¹ (HL); samples were taken 1 h (RNA) or 4 h (protein) after exposure to HL. **a, c.** Accumulation of *LHCSR3* mRNA at the indicated conditions normalized to WT LL ctrl (n = 3 biological samples, mean ± s.d.). The p-values for the comparisons of acetate and $CO_2$ conditions to air are based on ANOVA Dunnett's multiple comparisons test of log10 transformed mRNA data as indicated in the graphs (*$P < 0.005$, **$P < 0.01$, ***$P < 0.001$, ****$P < 0.0001$, ns, not significant). Exact p-values can be found at the Source Data file. **b, d.** Immunoblot analyses of LHCSR3 and ATPB (loading control) under the indicated conditions. Representative datasets of experiments repeated three times.

acetate, suggesting that acetate metabolism resulted in higher intracellular $CO_2$ levels (Fig. 2b).

In another experiment, LL acclimated cells were shifted to HL (t = 0) and *LHCSR3* and *RHP1* transcript levels were assayed over a period of 9 h; acetate or high $CO_2$ were introduced 1 h after the shift to HL (Fig. 2d, e, note the 1 h time point highlighted in green on the x-axis). In agreement with Fig. 1 (HL panel), *LHCSR3* transcript accumulation increased by two orders of magnitude after 1 h exposure to HL in both WT and the *icl* mutant (Fig. 2d), while *RHP1* transcripts rapidly decreased (Fig. 2e), which likely resulted from a reduction in the concentration of intracellular $CO_2$ as a consequence of enhanced photosynthetic $CO_2$ fixation in the HL. Introduction of acetate or $CO_2$ to the cultures caused a rapid reduction in the level of *LHCSR3* expression in WT (Fig. 2d), with the decline much more pronounced with $CO_2$ supplementation. Supplementation with $CO_2$ or acetate also caused an increase of *RHP1* transcript relative to the control. In contrast, in the *icl* mutant, the decline in the level of the *LHCSR3* transcript and the increase in the level of the *RHP1* transcript was the same in cells with and without acetate supplementation, while the effect of $CO_2$ was similar to that of WT cells (Fig. 2d, e). Furthermore, WT cells consumed about half of the acetate in the medium over the course of the experiment, while none of the acetate was consumed by the *icl* mutant (Fig. 2f). These results strongly suggest that $CO_2$ inhibits the accumulation of the *LHCSR3* transcript and that the decline of *LHCSR3* mRNA in WT cells supplemented with acetate is a consequence of the $CO_2$ released as the acetate is metabolized. The extent of this inhibition by acetate-derived $CO_2$ appears to depend mostly on the rate of photosynthetic $CO_2$ fixation (consumption of $CO_2$) because acetate was taken up by WT cells at similar rates under both LL and HL conditions (Fig. 2c, f). Indeed, under LL conditions, where $CO_2$ fixation is slow, acetate and $CO_2$ repressed *LHCSR3* to the same extent (Fig. 2a); under HL conditions, where $CO_2$ fixation is much faster, the effect of acetate on the *LHCSR3* transcript level was much smaller than that of $CO_2$, which was continuously provided in excess (5%) via sparging (Fig. 2d).

We also employed constraint-based metabolic modelling to assess in silico whether acetate metabolism in *Chlamydomonas* leads to an increase in the concentration of intracellular $CO_2$ under different growth conditions (Supplementary Note 1, Supplementary Fig. 2, Supplementary Tables 1–3, Supplementary Data 1–3). The findings from this approach support the hypothesis that there are changes in the internal $CO_2$ concentration under autotrophic and mixotrophic growth conditions at different light intensities. These predicted changes in internal $CO_2$ levels under the different conditions for the WT and mutant cells are congruent with the levels of accumulation of *LHCSR3* transcripts that were measured.

## CIA5 links HL and low $CO_2$ responses

The responses to HL and low $CO_2$ have been traditionally studied separately, despite several lines of evidence suggesting that they are integrated[26,32]. To elucidate the molecular connection between photoprotection and CCM, we analyzed mRNA accumulation of twelve genes implicated as functionally involved in the CCM, previously shown to be strongly expressed under low $CO_2$ conditions[33,34] and/or to be under the control of CIA5[26]. Specifically, we analysed twelve CCM-related genes encoding LOW-$CO_2$-INDUCIBLE PROTEIN B (LCIB) and E (LCIE), involved in $CO_2$ uptake; HIGH-LIGHT ACTIVATED 3 (HLA3), LOW $CO_2$-INDUCED 1 (LCI1), CHLOROPLAST CARRIER PROTEIN 1 (CCP1), CCP2, LCIA, BESTROPHINE-LIKE PROTEIN 1 (BST1), acting as Ci transporters; carbonic anhydrases CAH1, CAH3, CAH4; the nuclear regulator LOW-$CO_2$-STRESS RESPONSE 1 (LCR1).

When LL-acclimated, air-sparged WT, *icl* and *icl-C* strains were exposed to HL (experiment described in Fig. 1) a marked increase (5 to 600-fold) in CCM transcript levels was observed in WT cells (Supplementary Fig. 3), in accordance with recent studies;[28,32] this increase was strongly suppressed by $CO_2$ and to a lesser extent by acetate, which did not affect CCM gene expression in the *icl* mutant (Supplementary Fig. 3). This pattern of mRNA accumulation was essentially identical to that of *LHCSR3* (Fig. 1a), highlighting the tight connection between HL

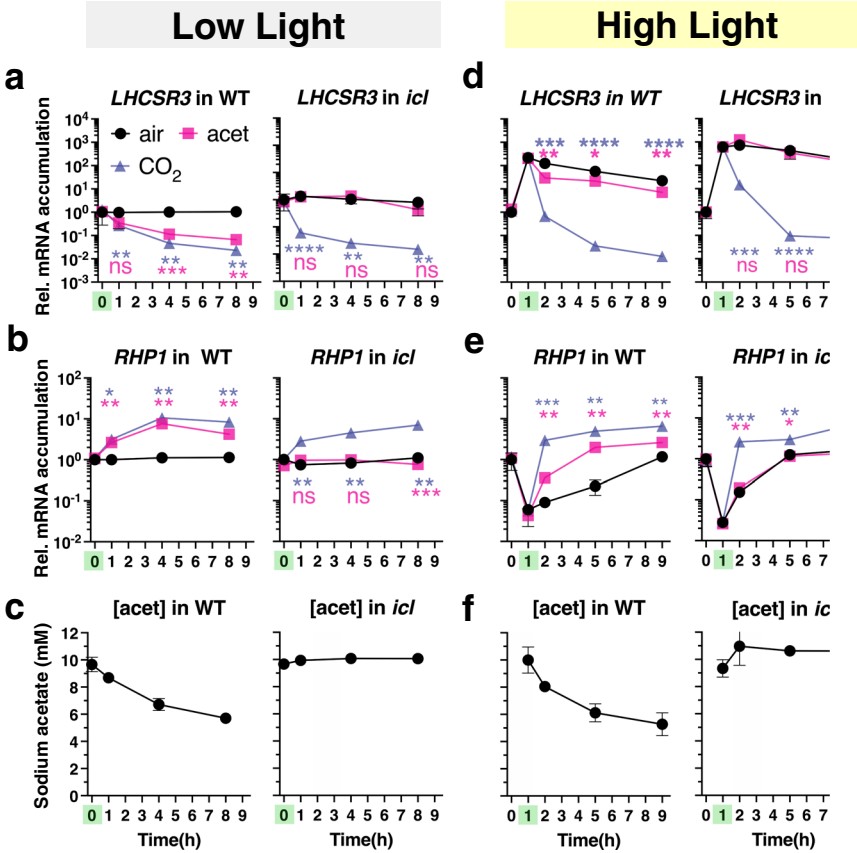

**Fig. 2 | LHCSR3 inhibition is driven by CO₂ derived from the metabolism of acetate.** Experiment at LL: **a**, **b** mRNA accumulation of *LHCSR3* and *RHP1* and **c** concentration of sodium acetate in the growth medium in WT and *icl* strains. Cells were acclimated overnight at LL (15 μmol photons m⁻² s⁻¹) in HSM sparged with air. At t = 0 cells either continued being sparged with air (labelled "air"); or sparged with air and supplemented with 10 mM sodium acetate (labelled "acet"); or sparged with air enriched with 5% CO₂ (labelled "CO₂"). The addition of acetate or CO₂ is indicated with a green mark on the x-axis. Samples were taken at t = 0, 1 h, 4 h and 8 h. Experiment at HL: **d**, **e** mRNA accumulation of *LHCSR3* and *RHP1* and **f** concentration of sodium acetate in the growth medium in WT and *icl* strains. Cells were acclimated overnight at LL (15 μmol photons m⁻² s⁻¹) in HSM sparged with air; at t = 0 light intensity was increased to 600 μmol photons m⁻² s⁻¹.

At t = 1 h cells either continued being sparged with air (labelled "air"); or sparged with air and supplemented with 10 mM sodium acetate (labelled "acet"); or bubbled with air enriched with 5% CO₂ (labelled "CO₂"), always at 600 μmol photons m⁻² s⁻¹. The time of addition of acetate or CO₂ is highlighted in green on the x-axis. Samples were taken at t = 0, 1 h, 2 h, 5 h and 9 h. (*n* = 3 biological samples, mean ± s.d.). The *p* values for the comparisons of acetate and CO₂ conditions to air (LL; t = 1, 4, 8 h, HL; t = 2, 5, 9 h) are based on ANOVA Dunnett's multiple comparisons test of log10 transformed mRNA data as indicated in the graphs (*P < 0.005, **P < 0.01, ***P < 0.001, ****P < 0.0001, ns, not significant), following the color-code of the datasets. Exact p-values can be found at the Source Data file. Please note that in some cases the error bars are smaller than the data point symbols.

and low-CO₂ responses in *Chlamydomonas*. The CO₂-mediated repression was more pronounced for most of the CCM genes relative to *LHCSR3* (Fig. 1a, c, Supplementary Fig. 3).

CIA5 has been shown to regulate the accumulation of transcripts from both the CCM genes[26,27] and *LHCSR3*[28]. To obtain a comprehensive view of the photoprotection capacity of the *cia5* mutant, air-sparged WT and *cia5* cells grown in LL were shifted to HL, and the transcript and protein levels from the qE effector genes were monitored. Remarkably, a lack of *CIA5* resulted in much lower accumulation of *LHCSR3* mRNA than in WT cells; 50 times lower at LL and over 200 times lower at HL. This phenotype was fully reversed by ectopic expression of the WT *CIA5* gene (Fig. 3a). *PSBS* also showed a significant CIA5-dependent control at the mRNA level, although at a smaller extent (Fig. 3a). The *cia5* mutant accumulated slightly more *LHCSR1* mRNA in both LL and HL (~2 fold), however, this phenotype was not restored in the complemented *cia5-C* strain (Fig. 3a); we conclude that *LHCSR1* mRNA accumulation is CIA5-independent. We also quantified the accumulation of mRNAs of *CAH1* and *LCIA*, which are known to be strongly dependent on CIA5[27,35]. As expected, the *cia5* mutant cells failed to activate either of those genes in HL while their activation was fully restored in the complemented *cia5-C* strain (Fig. 3a).

At the protein level, no LHCSR3 protein was detected in the *cia5* mutant in either LL or HL (Fig. 3b). We were unable to immunologically detect the PSBS protein under these experimental conditions, in agreement with previous findings showing that PSBS protein accumulation is highly transient in cell cultures bubbled with air[6]. Importantly, the LHCSR1 protein accumulated to high levels in the mutant under both LL (conditions in which no protein is apparent in the WT) and HL conditions; this phenotype was fully reversed by ectopic expression of the WT *CIA5* gene (Fig. 3b). This result suggests that CIA5 acts as a suppressor of LHCSR1 translation (and/or decreases protein stability) in both LL and HL. Our data additionally suggest that accumulation of LHCSR1 protein occurs through a compensatory, CIA5-controlled posttranscriptional mechanism that provides photoprotection under conditions in which the cells have almost no LHCSR3 protein (compare LHCSR1 and LHCSR3 immunoblots in Fig. 3b). Supporting this idea, the qE levels in *cia5*, although lower than WT and *cia5-C* (Fig. 3c and Supplementary Fig. 4), they were unexpectedly high considering the absence of LHCSR3 protein (Fig. 3b); we attribute this result to overaccumulation of LHCSR1 in this mutant (Fig. 3b). Together, our results demonstrate a key role of CIA5 in regulating photoprotection, activating *LHCSR3* and to a lesser extent *PSBS* transcription and suppressing LHCSR1 protein accumulation.

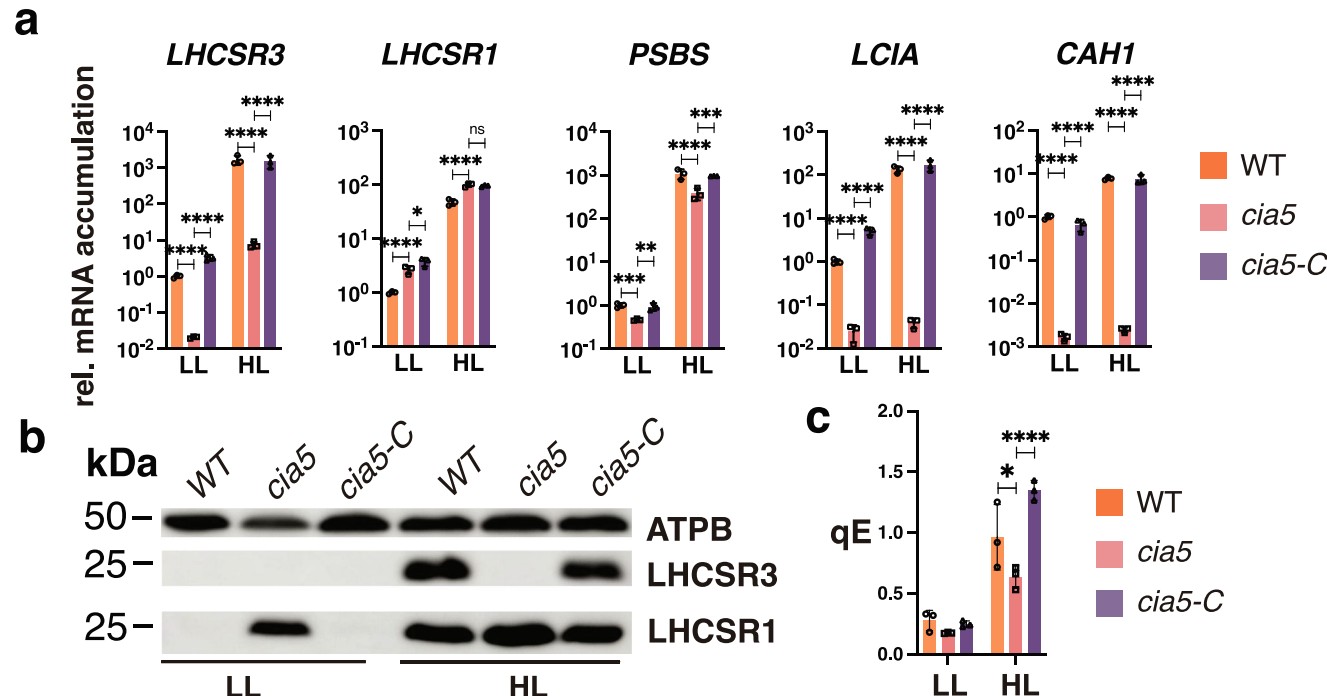

**Fig. 3 | Cross-talk of responses to HL and low-CO₂. a** CC-125 WT, *cia5* and *cia5*-c strains were acclimated for 16 h in LL (15 μmol photons m⁻² s⁻¹) in HSM bubbled with air (labelled as "LL"); after sampling for the LL conditions, light intensity was increased to 600 μmol photons m⁻² s⁻¹ (HL); samples were taken after 1 h (RNA) and 4 h (protein and photosynthesis measurements). Accumulation of mRNA of genes at the indicated conditions were normalized to WT LL ctrl. (*n* = 3 biological samples, mean ± s.d.). The p-values for the comparisons of WT with *cia5* and *cia5* with *cia5-C* are based on ANOVA Dunnett's multiple comparisons test of log10 transformed mRNA data as indicated in the graphs. **b** Immunoblot analyses of LHCSR3, LHCSR1 and ATPB (loading control) under the indicated conditions; PSBS was non-detectable at these experimental conditions. Representative dataset of experiment repeated three times. **c** qE of WT, *cia5* and *cia5-C* under LL and HL conditions (*n* = 3 biological samples, mean ± s.d.). The statistical analyses (two-way ANOVA Tukey's multiple comparison test) are shown in the graph. Exact p-values can be found at the Source Data file. Raw fluorescence and NPQ curves can be seen in Supplementary Fig. 4.

## CIA5 and CO₂ availability regulate LHCSR1 protein stability

The high levels of accumulation of LHCSR1 protein in the *cia5* mutant (Fig. 3b) suggest that CO₂ availability could be the key determinant for LHCSR1 protein accumulation, as CIA5 is not functional under high CO₂ levels[15–17]. Given the novelty of this finding, we decided to perform additional experiments to provide more details concerning LHCSR1 regulation. LL-acclimated WT cells sparged with air were exposed to HL sparged with air or 5% CO₂ and the mRNA and protein levels were quantified over a 25-h period. Upon initial exposure to HL, *LHCSR1* mRNA rapidly increased (2 orders of magnitude in 1 h) and then decreased to the initial level (between 4 and 8 h), in agreement with a previous report[36], in the presence or absence of high CO₂ (Fig. 4a). In contrast, the presence of high CO₂ sustained high levels of LHCSR1 protein over the 25-h incubation period relative to cultures sparged with air (Fig. 4b). These results suggest that elevated CO₂ either promotes translation of *LHCSR1* mRNA or is involved in stabilizing the protein once it is synthesized. This contrasts with the behaviour of LHCSR3 for which there was a strong correlation between the level of mRNA and protein (the RNA was 3 orders of magnitude lower in 5% CO₂ and the protein was no longer detected) (Fig. 4). The kinetics of *PSBS* transcript accumulation in HL very much resembled those of *LHCSR1*, with CO₂ not having a strong impact on transcript accumulation (Fig. 4a). PSBS protein accumulation was not detectable under the experimental conditions used. Taken together, our data demonstrate the critical importance of CIA5 and CO₂ in regulating the different qE effectors, mainly *LHCSR3* and less strongly *PSBS* at the transcript level, and LHCSR1 at the protein level.

## Intracellular CO₂ levels regulate photoprotective and CCM gene expression in the absence of light

To de-convolute the light and CO₂ signals regulating *LHCSR3*, we exposed the cells to different light intensities and CO₂ concentrations (Supplementary Fig. 5). High CO₂ levels completely abolished the accumulation of LHCSR3 protein at all light intensities, in accord with the results of Fig. 1b, d and Fig. 4b. On the contrary, low CO₂ levels led to very high accumulation of LHCSR3 protein at 150 and 300 μmol photons m⁻² s⁻¹. Under low CO₂, LHCSR3 protein was also detectable even at the very low light intensity of 10 μmol photons m⁻² s⁻¹ (Supplementary Fig. 5), as previously demonstrated[6].

Prompted by this result, we tested whether changes in CO₂ levels could activate transcription of *LHCSR3* in complete darkness. We shifted air-sparged cells to sparging with CO₂-free air (Very low CO2; VLCO₂) in complete darkness and to our surprise, we observed that despite the absence of light, that a drop in CO₂ availability was sufficient to trigger *LHCSR3* mRNA accumulation by ~ 700-fold (Fig. 5a), with an increase in accumulation of the protein by 3-fold (Fig. 5b, c; compare WT air with WT VLCO₂). In addition, when HL was combined with VLCO₂, which is expected to result in an even greater reduction in the intracellular CO₂ concentration, the levels of *LHCSR3* mRNA and protein further increased, reaching levels of ~4500-fold (mRNA) and 21-fold (protein) compared to air dark conditions (Fig. 5a-c). Interestingly, this light-independent regulation of mRNA accumulation was under the control of CIA5 as the accumulation of *LHCSR3* transcripts was abolished in the *cia5* mutant (Fig. 5a) and a full reversal of these phenotypes (gene expression and protein levels) was observed in the *cia5-C* strain (Fig. 5a, b). We also observed significant *LHCSR3* transcript accumulation in the *cia5* mutant when cells were shifted from dark-air to HL-VLCO₂, which was, however, 9-fold lower compared to the WT (Fig. 5a), and that was rescued to WT-levels in the *cia5-C* complemented line. This CIA5-independent regulation of mRNA in the presence of light could account for the contribution of light signaling in *LHCSR3* gene expression, possibly via phototropin[10] or via the generation of reactive oxygen species[28].

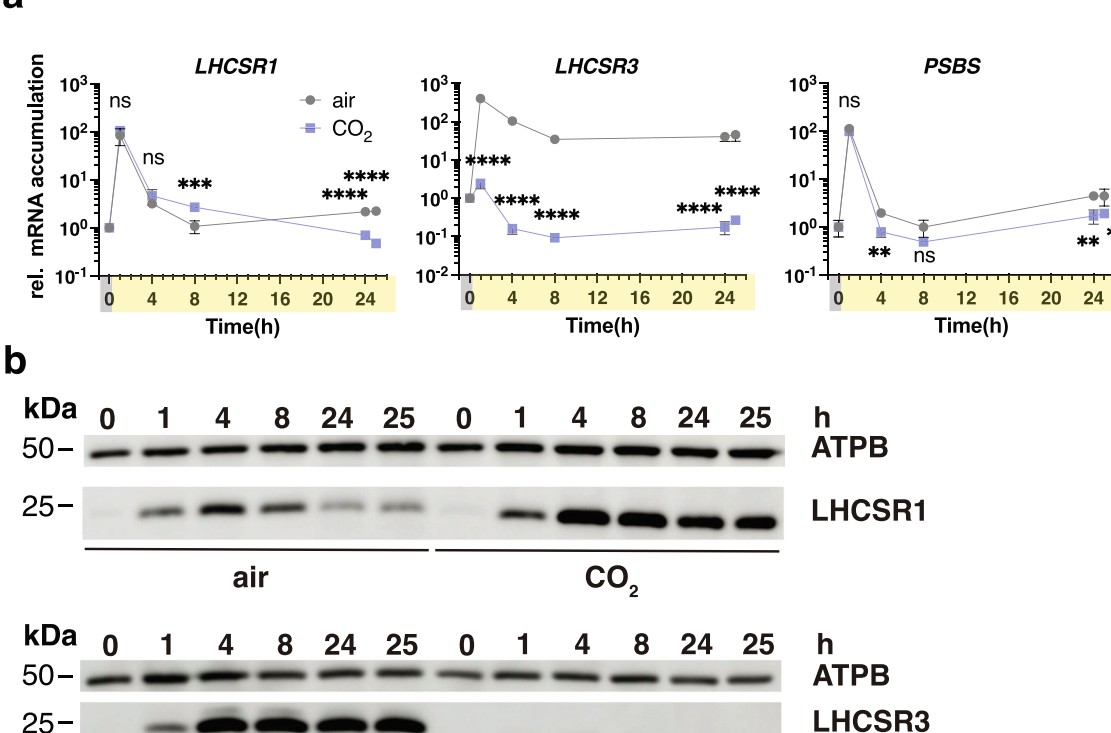

**Fig. 4 | Kinetic resolution of photoprotective gene and protein expression at different light and $CO_2$ availabilities.** Cells were acclimated overnight at LL (15 μmol photons $m^{-2} s^{-1}$) bubbled with air (labelled "air"). At t = 0 the light intensity was raised to 600 μmol photons $m^{-2} s^{-1}$ under air bubbling or bubbling with 5% $CO_2$ and mRNA and protein were followed for 25 h. **a** *LHCSR1, LHCSR3* and *PSBS* mRNA accumulation. (n = 3 biological samples, mean ± s.d.). The p-values for the comparisons of $CO_2$ conditions to air for t = 1, 4, 8, 24 and 25 h are based on two-way ANOVA Šídák's multiple comparisons test of log10 transformed mRNA data as indicated in the graphs (*P < 0.005, **P < 0.01, ***P < 0.001, ****P < 0.0001, ns, not significant). Exact p-values can be found at the Source Data file. **b** Immunoblot analyses of LHCSR1, LHCSR3 and ATPB (loading control). Representative dataset of experiment repeated three times.

We could observe that *LHCSR1* transcripts were also induced in the dark (shift from dark-air to dark-VLCO₂), but this induction was very low (7-fold) and appeared to be CIA5 independent (Supplementary Fig. 6a). At the protein level however, LHCSR1 over-accumulated in the *cia5* mutant under all conditions tested (Supplementary Fig. 6b), confirming our previous findings (Fig. 3b). *PSBS* also showed a CIA5-dependent dark induction of transcripts (shift from dark-air to dark-VLCO₂), although this induction was low (5-fold); complementation with the CIA5 gene (*cia5-C* strain) did not rescue the phenotype in dark-air conditions and only partially rescued it under dark-VLCO₂ (Supplementary Fig. 6a, b). Both mRNA and protein accumulation of PSBS accumulated in a CIA5-dependent manner when cells were shifted from dark-air to HL-VLCO₂ (Supplementary Fig. 6a, b); under these conditions the phenotypes were fully reversed in the *cia5-C* strain. This CIA5-dependent regulation of PSBS can most likely explain previously reported findings that PSBS protein accumulation was responsive to $CO_2$ abundance, with its accumulation reaching maximum levels under low $CO_2$ and HL conditions[6].

We also measured CCM-related gene expression in the dark. As shown in Fig. 5a and Supplementary Fig. 6a, high levels of CCM-related transcripts were observed in the dark when the cells experienced VLCO₂ conditions (compare "dark air" with "dark VLCO₂"). The combination of HL and VLCO₂ conditions, either elicited very small (less than two-fold) or no additional increase (compare "HL VLCO₂" to "dark VLCO₂") in their level of the mRNA accumulation (Fig. 5a and Supplementary Fig. 6a). As expected CIA5 was critical for expression of the CCM genes under all conditions tested (Fig. 5a and Supplementary Fig. 6a).

Our data points out that the LHCSR1 protein overaccumulation in *cia5* was fully reversed only when *cia5-C* cells were pre-acclimated in the light (Fig. 3b); when pre-acclimation took place in the dark the phenotype was only partially rescued (Supplementary Fig. 6b). The same is true for the mRNA accumulation of *PSBS* (compare LL; Fig. 3a with air-dark; Supplementary Fig. 6a), *CAH1* (compare LL; Fig. 3a with air-dark; Supplementary Fig. 6a), *LCIA* (compare LL; Fig. 3a with air-dark; Fig. 5a), while in the case of LHCSR3 a full reversal of the CIA5-dependent phenotype was seen no matter what pre-acclimation strategy was followed (Figs. 3a, b and 5a, b). A plausible explanation for these results is the differential accumulation of CIA5 protein in the different acclimation regimes due to the promoter used. In line with this explanation, CIA5 expression in *cia5-C* is driven by the light-inducible promoter of the *PSAD* gene, and, as a result, less CIA5 protein accumulated in the dark-acclimated *cia5-C* compared to the LL-acclimated (Supplementary Fig. 7a). This in turn affects the relative abundance of CIA5 available for binding with its target molecules (DNA binding sites or CIA5-interacting proteins), ultimately affecting the reversal of the CIA5-related phenotypes.

Overall, these data challenge the view concerning the regulation of photoprotection and CCM and bring $CO_2$ to the forefront as a crucial signal controlling *LHCSR3* and CCM-related genes induction in the absence of light.

## Link between photosynthetic electron transfer and $CO_2$ intracellular concentration

Our finding that *LHCSR3* is regulated by light-independent $CO_2$ availability has guided us in revising the way in which we view the impact of

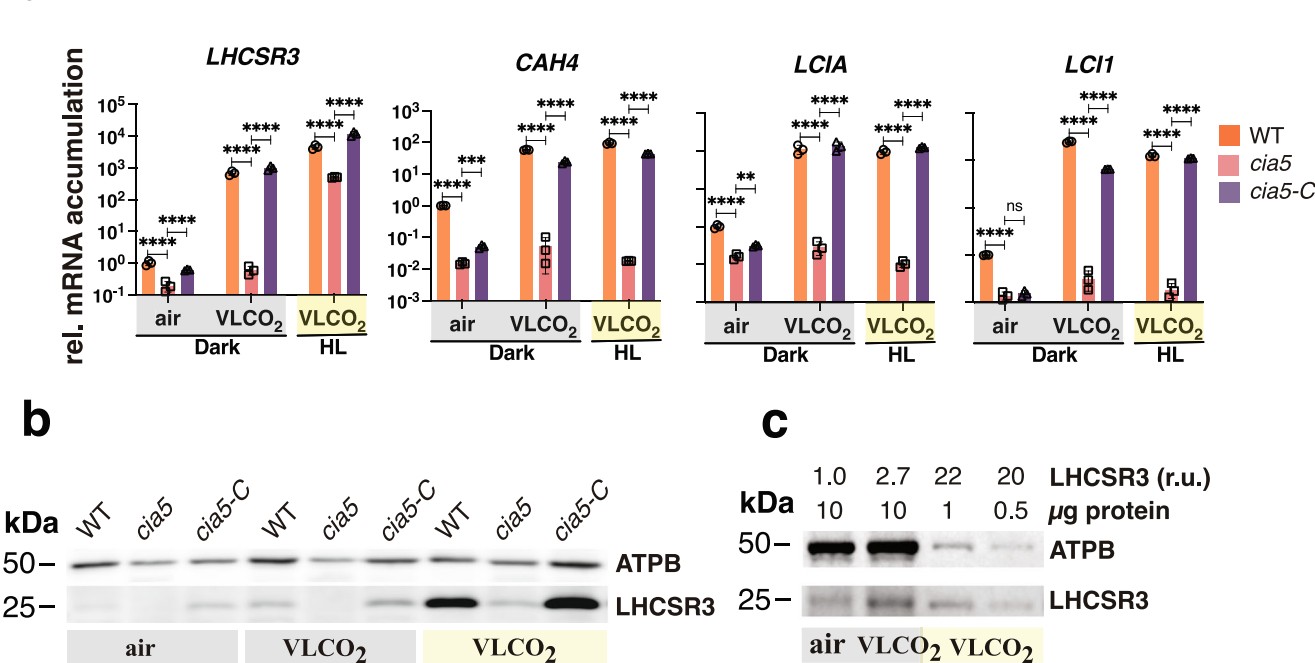

**Fig. 5 | Low CO₂ levels can trigger qE and CCM genes in the absence of light.** WT, *cia5* and *cia5-C* cells were bubbled with air overnight in darkness; next day air bubbling was either maintained or replaced by $CO_2$-limited-air bubbling in the darkness or in the presence of 600 μmol photons m⁻² s⁻¹ light. Sampling was performed after 1 h (RNA) or 4 h (protein). **a** mRNA accumulation of *LHCSR3.1* (qE gene) and *CAH4, LCIA, LCI1* (CCM genes) in WT, *cia5* and *cia5-C*. Data were normalized to WT air dark; (*n* = 3 biological samples, mean ± s.d.). The p-values for the comparisons of WT with *cia5* and *cia5* with *cia5-C* are based on ANOVA Dunnett's multiple comparisons test of log10 transformed mRNA data as indicated in the graphs (*$P < 0.005$, **$P < 0.01$, ***$P < 0.001$, ****$P < 0.0001$, ns, not significant). Exact p-values can be found at the Source Data file. **b** Immunoblot analyses of LHCSR3 and ATPB (loading control) under the indicated conditions. Representative dataset of experiment repeated three times. **c** Immunoblot analyses of LHCSR3 and ATPB (loading control) of WT samples presented in **b**. Above the immunoblot shown are the amount of protein loaded per lane and the quantification of LHCSR3 protein accumulation (calculated as LHCSR3 /ATPB ratio) normalized to the air dark conditions. Representative dataset of experiment repeated three times.

photosynthetic electron transfer (PET) on LHCSR3 accumulation; i.e. inhibition of LHCSR3 accumulation in photosynthetic mutants or WT cells treated with photosynthetic inhibitors[10,11].

We propose that enhanced PET, occurring under HL conditions, facilitates $CO_2$ fixation, draws down the intracellular $CO_2$ concentration and induces *LHCSR3* transcription. In contrast, when PET is impaired, intracellular $CO_2$ levels increase, thus promoting *LHCSR3* transcript inhibition. In accordance, there is a marked increase of $CO_2$ in cultures treated with DCMU, an inhibitor of photosystem II[37], measured either as dissolved $CO_2$ in the culture medium[30], or as $CO_2$ in the air stream coming from the headspace of the column bioreactor (Fig. 6a). In order to test our hypothesis, we analyzed the combined effect of DCMU and $CO_2$ on the accumulation of mRNA from the *LHCSR3* and two CCM genes in WT cultures shaken without or with VLCO₂ sparging. In accord with previous reports[10,38], DCMU completely blocked the HL elicited accumulation of *LHCSR3* mRNA; *LHCSR3* mRNA after 1 h exposure to HL diminished to ten times lower levels than the initial LL levels (shown as dotted line in graph) (Fig. 6b), which most likely reflects the degradation of the transcripts following inactivation of the gene after the addition of DCMU. Previous work has shown that *LHCSR3* transcripts are rapidly lost once the gene becomes inactive[10] which has also been observed for the *CAH4* transcript[39]. However, when the cultures were sparged with VLCO₂ air, which would result in the maintenance of a continuous VLCO₂ concentration in the cultures, a large part of the DCMU elicited inhibition was relieved (Fig. 6b), supporting the idea that light primarily impacts *LHCSR3* transcript levels by altering $CO_2$ consumption and the intracellular (and/or extracellular) $CO_2$ concentration. In contrast to *LHCSR3*,

sparging with VLCO₂ only partly relieved the suppression of transcript accumulation for the CCM genes in the presence of DCMU (Fig. 6b). This difference may reflect the fact that CCM gene expression is solely regulated by $CO_2$ via CIA5 (Fig. 5a) and that sparging with VLCO₂ in the presence of DCMU does not reduce the $CO_2$ levels enough to attain full gene activation. It is also possible that longer incubation time with VLCO₂ would have relieved a larger part of the DCMU-elicited inhibition of CCM genes (Fig. 6b) as implied by the slow kinetics of *CAH4/5* mRNA accumulation when cells are shifted from 5% $CO_2$ to air[39].

## Discussion

In this work, we presented findings that advance our understanding of integration between $CO_2$- and light-dependent signaling in *Chlamydomonas*. We propose that the intracellular level of $CO_2$, defined by the equilibrium between light-driven $CO_2$ fixation in chloroplasts and the generation of $CO_2$ by mitochondrial metabolism (e.g. acetate assimilation), is a key regulator of two major processes in photosynthetic organisms: the CCM and photoprotection (Fig. 7).

To better understand the role of $CO_2$ in regulating photoprotection and its integration with light, we designed experiments to separate the effects of the two signals (Figs. 5–7); we reduced the concentration of $CO_2$ in the microalgae medium by sparging it with VLCO₂ in complete darkness. This abrupt change in $CO_2$ levels experienced by the cultures in the dark may be considered a condition only encountered in the laboratory. However, in certain ecological niches, such as soil or catchments with elevated levels of organic matter[33], *Chlamydomonas* would encounter changes in the levels of $CO_2$ that would be dependent on the microbes and the ratio between

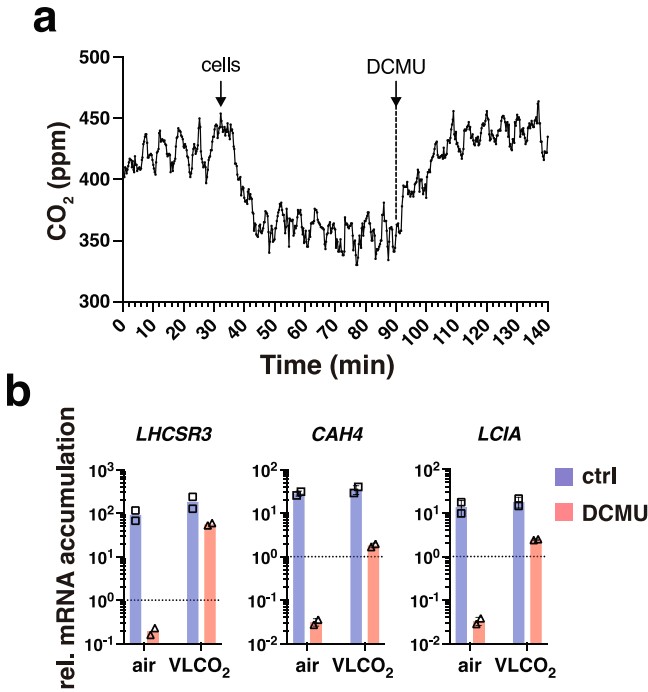

**Fig. 6 | Photosynthetic electron transfer draws down the intracellular $CO_2$ concentration, relieving inhibition of *LHCSR3* transcription. a** $CO_2$ concentration measured in the air stream coming out of the headspace of a column filled with 50 mL of HSM, sparged with air under HL. The two arrows in the graph indicate the addition of cells or DCMU. **b** WT cells were acclimated to LL HSM overnight shaken in flasks; the next day they were exposed to 300 μmol photons m$^{-2}$ s$^{-1}$ light in the presence or absence of 40 μM DCMU, shaken in flasks without or with sparging with VLCO$_2$. Samples were taken after 1 h. Presented are mRNA accumulation of *LHCSR3*, *CAH4*, *LCIA*. Data were normalized to LL (shown as dotted line in graph); n = 2 biological samples.

respiration and photosynthesis in the environment. Our experimental setup allowed us to observe a strong increase of *LHCSR3* transcript levels when cells were shifted from air-$CO_2$ to VLCO$_2$ levels in the dark (Fig. 5), a very surprising result as the accumulation of *LHCSR3* mRNA was considered so far to be strictly light-dependent[5,11,38]. Moreover, with this strategy we can disentangle light from $CO_2$ signalling effects; while dark induction of *LHCSR3* under $CO_2$-depletion was completely dependent on CIA5, light could still strongly impact expression of all photoprotective genes in the *cia5* mutant, which was not the case for CCM gene expression that was completely abolished in the light or dark in the absence of CIA5 (Fig. 5 and Supplementary Fig. 6). This impact of light on qE gene expression may be the consequence of photoperception (e.g. PHOT1)[10], but also the generation of light-dependent signals such as reactive oxygen species[28]. Furthermore, a CIA5-independent regulation (also observed in Fig. 3a) explains *LHCSR3* induction in high $CO_2$-acclimated WT cells (cells in which CIA5 is not functional[15–17]) as they transition from LL to HL (Fig. 1a), which was not observed for CCM genes tested under identical conditions (Supplementary Fig. 3); it also explains why the $CO_2$-mediated repression was more pronounced for most of the CCM genes relative to LHCSR3 (Fig. 1a, c, Supplementary Fig. 3). $CO_2$ and CIA5 appear to be of paramount importance in signal integration and transduction, regulating expression of both photoprotection and CCM genes. For instance, $CO_2$ represses the UV-B elicited, UVR8-mediated expression of *LHCSR3*, and CIA5 is absolutely required for this expression[28]. Moreover, our results have shown that high $CO_2$ levels or the absence of CIA5 have a severe impact on *LHCSR3* gene expression and, although HL can still induce *LHCSR3* transcription, no protein is detected (Figs. 1, 3 and 5).

Besides transcriptionally controlling *LHCSR3*, CIA5 post-transcriptionally controls LHCSR1. Our view on LHCSR1 regulation by light and CIA5 is as follows: under LL conditions, LHCSR1 protein accumulates in *cia5* while it is non-detectable in WT and *cia5-C* (Fig. 3b), suggesting that CIA5 suppresses LHCSR1 protein accumulation. Exposure to HL triggers a CIA5-independent *LHCSR1* mRNA accumulation (Fig. 3a), possibly driven by reactive species, previously shown to favor *LHCSR1* mRNA accumulation[40]. As a result, LHCSR1 protein accumulates in WT and *cia5-C* in HL, despite the fact that suppression of LHCSR1 protein by CIA5 still occurs; indeed, LHCSR1 accumulates to higher levels in the *cia5* mutant as compared to WT and *cia5-C* under HL conditions (Fig. 3b). In line with the above observations in the *cia5* mutant, high levels of LHCSR1 protein accumulate in WT under high $CO_2$, conditions that inactivate CIA5 (Fig. 4b). Put together, our findings unveil a multilevel role of CIA5 in regulating qE; inactivation of CIA5 in high $CO_2$ or by eliminating the CIA5 gene blocks *LHCSR3* transcript accumulation, while it promotes LHCSR1 protein accumulation (Figs. 3, 4). Further investigation will be required to explain how a single nuclear factor, CIA5, can control cellular processes happening in different cellular compartments; transcription in the nucleus and translation in the cytosol.

Our results provide an interpretation of the findings that PET is required for LHCSR3 accumulation[11], activation of the CCM and expression of CCM genes[41]. We propose that $CO_2$, either provided directly or indirectly through metabolic generation, represents a critical link between PET and transcriptional regulation of *LHCSR3* and the CCM genes (Fig. 6). Photosynthesis draws down cellular $CO_2$ levels, and therefore, blocking photosynthesis with DCMU leads to the accumulation of $CO_2$ (Fig. 6a) which elicits *LHCSR3* repression, while sparging DCMU-treated cells with VLCO$_2$ almost fully derepresses *LHCSR3* (and partially CCM) expression (Fig. 6b). DCMU also upregulates genes of the leucine degradation pathway[42] leading to the generation of acetoacetate and acetyl-CoA, which can lead to oxidative $CO_2$ production. Whether leucine itself has a regulatory role or $CO_2$ is the key regulator deserves further attention. It is tempting to propose that $CO_2$ is a retrograde signal that readily diffuses through the cell and impacts nuclear gene expression, which would integrate both mitochondrial and chloroplastic metabolic activities.

The way in which *Chlamydomonas* senses $CO_2$ is not clear. Our data, i.e. accumulation of *LHCSR3* and CCM genes in the dark, exclude the possibility that a metabolite produced by photorespiration plays a major signalling role, as previously proposed[43]. $CO_2$ itself might also serve as the metabolite being recognized by a putative sensor that could be controlled by carbamylation, a $CO_2$-mediated post-translational modification that regulates, among others, the activation of Rubisco[44]. Furthermore, the large number of adenosine and guanylyl cyclases in *Chlamydomonas*[45] suggests that cyclic nucleotides play an important role in controlling various processes in this alga; these metabolites have been shown to be involved in mating[46], regulation of flagellar beating and phototaxis[47–49], in regulating inorganic nitrogen assimilation[50] and in restoring LHCSR3 accumulation in the absence of phototropin[10]. Cyclases have been shown to act as $CO_2$ sensors (as bicarbonate) in mammalian cells[51], making it plausible that they can also serve as sensors in *Chlamydomonas*. As cyclic nucleotide signalling and calcium are tightly linked[51], we anticipate an important role for calcium in $CO_2$ sensing; calcium signalling has already been shown to be involved in the regulation of both *LHCSR3* and CCM genes[11,52].

Overall, our work shows that the intracellular $CO_2$ level is the main factor in regulating CCM genes and *LHCSR3* in *Chlamydomonas* (Fig. 7). Exposure to HL increases the $CO_2$ fixation rate which causes a drop in intracellular $CO_2$ which, in turn, actives both photoprotection- and CCM-related genes. Depletion of $CO_2$ is sufficient to drive high expression levels of CCM genes and *LHCSR3* even in complete darkness. On the other hand, high $CO_2$ levels, either generated through

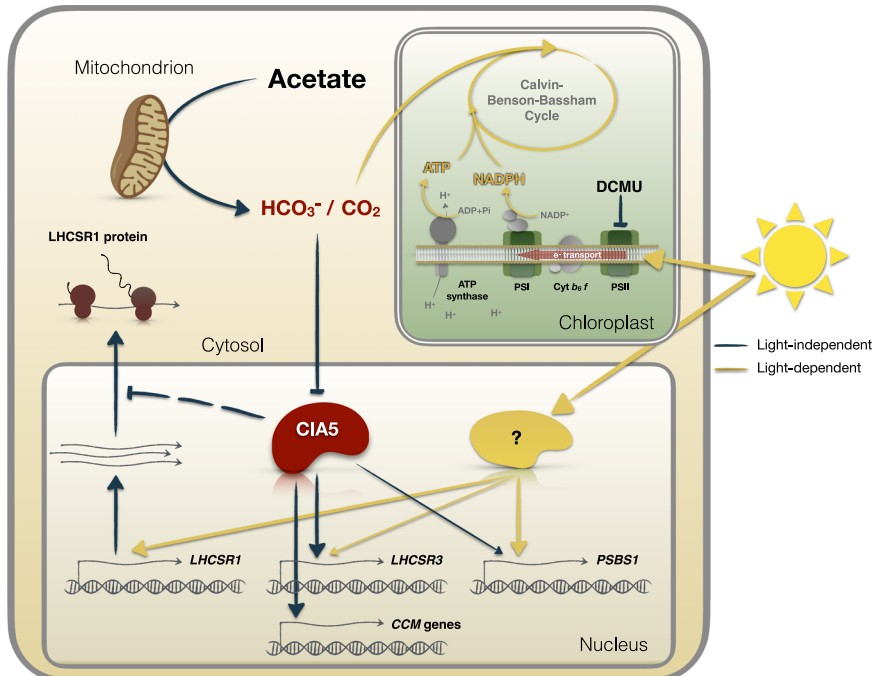

**Fig. 7 | CO₂- and light-dependent signals converge to regulate photoprotection and CCM in Chlamydomonas.** The intracellular levels of $CO_2$, defined by the equilibrium between $CO_2$ fixation in chloroplasts and the generation of $CO_2$ by mitochondrial metabolism (e.g. acetate assimilation) is the key determinant of the regulation of gene expression controlling two major processes of photosynthetic organisms: CCM and photoprotection. Changes in light availability have a direct impact on intracellular $CO_2$ levels; exposure to HL increases $CO_2$ fixation rates leading to depletion of $CO_2$ and to activation of not only photoprotection- but also CCM-related genes. Conversely, depletion of $CO_2$ is sufficient to drive high expression levels of CCM genes and *LHCSR3* even in complete darkness (indicated by the black arrows). High $CO_2$ levels, either exogenously supplied by sparging or metabolically produced via acetate metabolism or by inhibiting photosynthetic electron flow using DCMU, repress *LHCSR3* and CCM genes while at the same time they stabilize LHCSR1 protein levels. The close interconnection of photoprotection and CCM is further corroborated by the fact that CIA5, the regulator of expression of genes associated with the CCM, also exerts control over *LHCSR3* and to a lesser extent over *PSBS* mRNA levels and acts as repressor of LHCSR1 protein accumulation. Independent of CIA5, light strongly impacts expression of all of these photoprotective genes (yellow arrows). This impact can be the consequence of both photoperception (e.g. phototropin) and the production of reactive oxygen species.

enhanced respiratory activity or impaired photosynthetic electron transport, repress *LHCSR3* and CCM genes while at the same time stabilizing the LHCSR1 protein, which likely acts as a backup photoprotection protein under conditions where LHCSR3 is not expressed. Furthermore, our data reveals a closer interconnection of photoprotection and CCM as CIA5, the CCM master regulator, also exerts control over *LHCSR3* and to a lesser extent over *PSBS* mRNA levels, while repressing LHCSR1 protein accumulation. Our findings highlight the need to develop an integrated approach that examines the role of $CO_2$ and light, not only as substrates of photosynthetic $CO_2$ fixation, but also as signals regulating photoprotection, CCM, and at a wider context genome-wide gene expression.

## Methods
### Chemicals
DCMU (3-(3,4-dichlorophenyl)−1,1-dimethylurea) was purchased from Sigma. Stock solutions of DCMU were prepared in ethanol (40 mM).

### Strains and conditions
*Chlamydomonas* strains were grown under 20 μmol photons m⁻² s⁻¹ in Tris-acetate-phosphate (TAP) medium[53] at 23 °C in Erlenmeyer flasks shaken at 125 rpm. For all experiments cells were transferred to Sueoka's High Salt medium[54] supplemented when needed with 10 mM sodium acetate, at 2 million cells mL⁻¹ in 80 mL capacity columns, unless otherwise stated, sparged with air, air enriched with 5% $CO_2$, or very low $CO_2$ air (VLCO₂; generated by passing the air through soda lime) and exposed to light intensities as described in the text and figure legends. *Chlamydomonas* strain 137c mt+ was used as WT. The *icl* (defective in *ICL1*; gene ID: Cre06.g282800), *icl-C* (*icl* strain

complemented with the WT *ICL* gene), *dum11* (defective in defective in ubiquinol cytochrome c oxidoreductase of the respiratory complex III; geneID: CreMt.g000300) and *cia5* (defective in *CIA5*, aka CCM1; geneID: Cre02.g096300; Chlamydomonas Resource Centre strain CC-2702) mutants were previously generated[15,21,29]. For complementation of *cia5*, a 3.5-kbp genomic DNA fragment from CC-125 containing the *CIA5* coding region was amplified by PCR using Platinum™ SuperFi™ DNA Polymerase (Thermo Fisher Scientific) and primers gib-cia5-fw and gib-cia5-rev (Supplementary Table 4), gel purified and cloned into pLM005[55] by Gibson assembly[56] for expression under control of the *PSAD* promoter. Junctions and insertion were sequenced, and constructs were linearized by EcoRV before transformation into *cia5*. Eleven ng/kb of linearized plasmid[55] mixed with 400 μL of $1.0 \times 10^7$ cells mL⁻¹ were electroporated in a volume of 120 mL in a 2-mm-gap electro cuvette using a NEPA21 square-pulse electroporator (NEPAGENE, Japan). The electroporation parameters were set as follows: Poring Pulse (300 V; 8 ms length; 50 ms interval; one pulse; 40% decay rate; + Polarity), Transfer Pulse (20 V; 50 ms length; 50 ms interval; five pulses; 40% decay rate; +/- Polarity). Transformants were plated onto solid agar medium containing 10 μg/ml paromomycin and screened for fluorescence using a Tecan fluorescence microplate reader (Tecan Group Ltd., Switzerland). Parameters used were as follows: YFP (excitation 515/12 nm and emission 550/12 nm) and chlorophyll (excitation 440/9 nm and 680/20 nm). Transformants showing a high YFP/chlorophyll ratio were further analyzed by immunoblotting using anti-FLAG antibodies (Supplementary Fig. 7b). Among the transformants analyzed the *cia5-C-a1* (*cia5-C* throughout the text) was retained for further analyses in the present study, after verifying that it grows similarly with the WT under phototrophic conditions on agar (Supplementary

Fig. 7c). Unless otherwise stated, LL conditions corresponded to 20 $\mu$mol photons m$^{-2}$ s$^{-1}$ while HL conditions corresponded to 600 $\mu$mol photons m$^{-2}$ s$^{-1}$ of white light (Neptune L.E.D., France; see Supplementary Fig. 8 for light spectrum). All experiments were repeated three times to examine reproducibility, unless otherwise stated.

### Light acclimation experiments

Cells were acclimated overnight in High Salt Medium (HSM) in LL sparged with air, in the presence or absence of acetate, or sparged with 5% $CO_2$. Following this acclimation period, cells were transferred from LL to HL, with all other conditions identical to those of the acclimation period. Samples were collected after 1 h for RNA analyses and after 4 h for protein analysis and measurements of photosynthetic activity.

### Fluorescence-based measurements

Fluorescence-based photosynthetic parameters were measured with a pulse modulated amplitude fluorimeter (MAXI-IMAGING-PAM, Heinz-Waltz GmbH, Germany). Prior to the onset of the measurements, cells were acclimated to darkness for 15 min. Chlorophyll fluorescence was recorded under different intensities of actinic light; starting with measurements in the dark (indicated as D below the x-axis of the graphs), followed by measurements at 21 $\mu$mol photons m$^{-2}$ s$^{-1}$ (indicated as L1 below the x-axis of the graphs) and 336 $\mu$mol photons m$^{-2}$ s$^{-1}$ (indicated as L2 below the x-axis of the graphs) and finishing with measurements of fluorescence relaxation in the dark. The calculations of the different photosynthetic parameter was performed based on[57] as follows: The relative photosynthetic electron transfer rate (rETR) was calculated as $(Fm' - F)/Fm' \times I$; $F$ and $Fm'$ are the fluorescence yield in steady state light and after a saturating pulse in the actinic light, respectively; $I$ is the light irradiance in $\mu$mol photons m$^{-2}$ s$^{-1}$; NPQ was calculated as $(Fm - Fm')/Fm'$; $Fm$ is the maximal fluorescence yield in dark-adapted cells; the effective photochemical quantum yield of photosystem II was calculated as $Y(II) = (Fm'-F)/Fm'$; qE was estimated as the fraction of NPQ that is rapidly inducible in the light and reversible in the dark.

### mRNA quantification

Total RNA was extracted using the RNeasy Mini Kit (Qiagen) and treated with the RNase-Free DNase Set (Qiagen). 1 $\mu$g total RNA was reverse transcribed with oligo dT using Sensifast cDNA Synthesis kit (Meridian Bioscience, USA). qPCR reactions were performed and quantitated in a Bio-Rad CFX96 system using SsoAdvanced Universal SYBR Green Supermix (BioRad). The primers (0.3 $\mu$M) used for qPCR are listed in Supplementary Table 4. A gene encoding G protein subunit-like protein (GBLP)[58] was used as the endogenous control, and relative expression values relative to *GBLP* were calculated from three biological replicates, each of which contained three technical replicates.

### $CO_2$ measurements

$CO_2$ concentration was measured in the air stream coming from the headspace of a HSM or culture-containing column using the $CO_2$ Probe GMP251 connected to the MI70 data logger from Vaisala (Vantaa, Finland).

### Immunoblotting

Protein samples of whole cell extracts (0.5 $\mu$g chlorophyll or 10 $\mu$g protein) were loaded on 4-20% SDS-PAGE gels (Mini-PROTEAN TGX Precast Protein Gels, Bio-Rad) and blotted onto nitrocellulose membranes. Antisera against LHCSR1 (AS14 2819, 1:15000 dilution), LHCSR3 (AS14 2766, 1:15000 dilution), ATPB (AS05 085, 1:15000 dilution) were from Agrisera (Vännäs, Sweden); previously described was antisera against *C. reinhardtii* PSBS[6] (used at a dilution of 1:1000). ATPB was used as a loading control. An antirabbit horseradish peroxidase-conjugated antiserum was used for detection at

1:10000 dilution. Mouse monoclonal antibody against FLAG was purchased from Sigma-Aldrich (F3165, St. Louis, MO, USA) and was used at a dilution of 1:15000. An anti-mouse horseradish peroxidase-conjugated antiserum (Jackson Immuno Research Europe LTD) was used as a secondary antibody for 3xFLAG immunoblotting (1:10000 dilution). The blots were developed with ECL detection reagent, and images of the blots were obtained using a CCD imager (ChemiDoc MP System, Bio-Rad). For the densitometric quantification, data were normalized with ATPB.

### Statistical analyses

Statistical methods were not used to predetermine the sample size. The experiments were not randomized, and the investigators were not blinded to allocation during experimental procedures and data assessment. All statistical tests were performed using the computing environment Prism 9 (Graphpad Software, LLC), at a significance level of 0.05. In order to conform mRNA accumulation data to the distributional assumptions of Analysis of Variance (ANOVA), i.e. the residuals should be normally distributed and variances should be equal among groups, two-way ANOVA was performed with log-transformed data $Y = \log X$ where X is mRNA accumulation[59].

### Reporting summary

Further information on research design is available in the Nature Portfolio Reporting Summary linked to this article.

## Data availability

The source data underlying Figs. 1–6 and Supplementary Figures 1, 3, 4, 6, 7 are provided as a Source Data file. The Source Data file also includes the exact p-values for Figs. 1a, c, 2a, b, d, e, 3a, c, 4a, 5a, and Supplementary figures 1a, b, 3, 6a. All biological material described in this study is available upon request. Source data are provided with this paper.

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

## Acknowledgements

We are grateful to Dr. Konomi Fujimura-Kamada for performing experiments to validate the dark induction of LHCSR3 in the Minagawa lab; to Claire Remacle for sharing the *icl* and *icl-C* strains and Pierre Cardol for the *dum11* strain. We thank Eric Soupene for valuable insights in the RHP1 induction conditions. We thank Dimitra Karageorgou for performing preliminary experiments in the project and Gilles Curien for fruitful discussions on aspects of microalgae metabolism. We would like to thank the following agencies for funding: The Human Frontiers Science Program through the funding of the project RGP0046/2018 (DP, ARG, PR, ES-L, ZN, AK); the French National Research Agency in the framework of the Young Investigators program ANR-18-CE20-0006 through the funding of the project MetaboLight (DP); the French National Research Agency through the funding of the Grenoble Alliance for Integrated Structural & Cell Biology GRAL project ANR-17-EURE-0003 (DP, MAR-S, GV, YY); the French National Research Agency in the framework of the Investissements d'Avenir program ANR-15-IDEX-02, through the funding of the "Origin of Life" project of the Univ. Grenoble-Alpes (DP, YY); the Prestige Marie-Curie co-financing grant PRESTIGE-2017-1-0028 (MAR-S); the International Max Planck Research School "Primary Metabolism and Plant Growth" at the Max Planck Institute of Molecular Plant Physiology (MA, ZN); the European Union's Horizon 2020 research and innovation program under the Marie Sklodowska-Curie grant agreement no. 751039 (ES-L); the program 'Plan Propio UCO' from University of Cordoba, Spain for postdoctoral Support (ES-L); the Carnegie Institution for Science and the Department of Energy, DE-SC0019417 (ARG); the Marie Curie Initial Training Network Accliphot FP7-PEPOPLE-2012-ITN; 316427 (SF, GF, DP); the Japan Society for the Promotion of Science, JSPS, for the grants-in-Aid for Scientific Research, KAKENHI, 21H04778 and 21H05040 (JM) and the German Research Foundation DFG HI 739/9.2 (MH).

## Author contributions

Conceptualization: M.A.R.-S., ES.-L., P.R., M.H., Z.N., J.M., A.R.G., D.P.; Methodology: M.A.R.-S., Y.Y., G.V., P.R., E.S.-L., Z.N., D.P.; Investigation: M.A.R.-S., S.F., Y.Y., G.V., R.T., A.K., A.T., G.K., G.A., M.A., F.I.; Supervision: G.F., Z.N., J.M., A.R.G., D.P.; Writing—original draft: M.A.R.-S., D.P.; Writing—review & editing: all authors

## Competing interests

The authors declare no competing interests.
