## [Peer Review File · Nature Communications]

Light-independent regulation of algal photoprotection by CO₂ availabilityReviewer #1 (Remarks to the Author):

This study used several mutants involved CO₂ metabolism, e.g. the *icl* mutant and *dum11* mutant, to investigate the effects of CO₂ availability on photoprotection in *Chlamydomonas*. The author found that CO₂ availability has a significant role in regulating the expression of LHCSR1/3 and PSBS. Moreover, the CCM regulator CIA5 controls the expression of LHCSR3 and PSBS whereas inhibits LHCSR1 protein accumulation. This study demonstrates that CO₂ availability regulates non-photochemical quenching. This is a very exciting finding and extends our understanding of non-photochemical quenching. More importantly, this study shows how light signal and CO₂ signal converge to regulate photoprotection. The experiments were well performed. The conclusions were well supported by the data. In my opinion, the finding in this study is a breakthrough in photoprotection.

I have no other comments but suggest the author may emphasize the significance of CO₂ availability controlling photoprotection.

Reviewer #2 (Remarks to the Author):

The manuscript by Ruiz-Sola et al describes a light-independent regulation mechanism of photoprotection by intracellular CO₂ levels. They first show that acetate metabolism could inhibit LHCSR3, an essential component of photoprotection. Moreover, they show that photoprotection genes especially LHCSR3 is regulated by CCM regulator CIA5 and environmental CO₂ levels. This work sheds new light on the crosstalk between CO₂ and light response of *Chlamydomonas* algal cells. Here are some points that may improve the manuscript:

Major Points:

1. The biggest concern I have is that authors tried to connect CO₂ availability to photoprotection. However, there is no direct measurement of intracellular CO₂. Also, intracellular CO₂ especially their spatial distribution could be complicated due to the presence of CCM. It might be better to soften some statement, saying that photoprotection proteins are regulated by either acetate metabolism or CCM activities, which may converge to the hypothesis that intracellular CO₂ levels could have a direct impact in photoprotection. In addition, it would be helpful if authors could comment on what technical development might allow people to directly measure spatial-resolved intracellular CO₂ levels, and how it could advance our understanding of photoprotection regulated by CO₂.
2. Why in Fig. 1a, in *icl* complement in LL, acetate shows similar level of LHCSR3 mRNA as in air? Does the complementation strain has lower level of ICL compared to wildtype? Or it's LL so that difference might be small?
3. In some cases, the complement strains do not always recapitulate the phenotype of wildtype. For example *icl-C* in Fig. 1a in LL and *cia5-C* in fig 3c. Have you considered comparing the protein or mRNA level of the protein in WT and complement strains? Or try using the native promoter? Or do you have other assays like spot test to show the complementation actually works?
4. What is the physiological significance that other than light, CO₂ could also regulate photoprotection? Does it mean that the optimal light intensity will be set by CO₂ fixation capacity. Once above the optimal intensity, excessive light will be dissipated as heat.
5. Could the authors comment more on the relationship between LHCSR1 and LHCSR3, in the context of different CO₂ and light conditions. It seems to me that LHCSR3 is predominant in photoprotection function, longer lasting after light exposure, but much more sensitive to different CO₂ levels. On the contrary, LHCSR1 is more independent with CO₂ levels, and could serve as a backup photoprotection protein.

Minor points:

1. In the section "Link between photosynthetic electron transfer and CO₂ intracellular concentration", the full name of PET should be written at least once.
2. In Fig. 2a, it is hard to distinguish air and CO₂ based on color. Consider making the line thicker in the legend, or use colors that are more distinguishable. Also in the legend of Fig 2a, the font size of "air" is bigger than "acet"

Reviewer #3 (Remarks to the Author):

I have read the manuscript "Light-independent regulation of algal photoprotection by CO₂ availability" by Ruiz-Sola et al. in which the authors analysed the molecular mechanisms regulating the process of CO₂ concentrating mechanisms (CCM) and phosphoprotection in *Chlamydomonas reinhardtii*. To this aim, the authors have used genetic and mathematical modelling approaches and show (among other things) that inhibition of LHCSR3 accumulation and CCM activity by acetate is at the level of transcription and a consequence of metabolically produced CO₂.

My expertise in computational modelling, so my comments mostly refer to that part of the work.

The authors have used constraint-based metabolic modelling to assess whether there are changes in the internal concentration of CO₂ under different trophic conditions and at different light intensities. The analyses performed are standard and take advantage of already available functions from the cobra toolbox.

Overall the optimization problems are clearly defined and seem to have a well-grounded biological counterpart. The analysis could have provided more insights into this problem, particularly going at the single reaction-level.

First, they used FVA to extract those reactions whose steady state flux ranges did not overlap between the WT and modelled mutants. However, it is not clear how this set of reactions was identified. Were these reactions not overlapping at all in their allowed flux ranges to maintain the solution optimal or some tolerance was introduced? The authors provide a supplementary table with all the flux distributions obtained in their simulations but it is not straightforward to derive the criteria adopted in each circumstance.

They also assessed that set of reactions showing differences in all the simulated conditions are enriched with CO₂-producing reactions. It is not clear, however, whether the set of CO₂ producing reactions remain the same and their fluxes increase or, rather, other CO₂ producing reactions come into play (providing additional intracellular CO₂). Authors could have used the `computeFluxSplits` to compute relative contributions of fluxes to the net production and consumption of CO₂ and check the CO₂ metabolism at the single reaction level. Are there specific reactions that contribute more than the others to the CO₂ intracellular increase?

Also, it would be interesting to know how downstream pathways react to this increase in CO₂ availability as, for example, Calvin-Benson-Bassham cycle. Again, the information is already there but it appears under-investigated.

I am a bit surprised by the choice of following the expression of 12 selected CCM genes rather than performing a whole-cell transcriptomic analysis. I understand that the target of this work resides in the analysis of this specific cellular process but an -omic approach here would have allowed to provide a more general view of the problem, besides answering the specific question posed here by the author. Also, it would have probably been able to suggest other putative CIA5 targets and, as such, pave the way to further analyses.

AUTHOR REBUTTALS TO REVIEWERS' COMMENTS:

We thank the reviewers for their time in reviewing this manuscript and their positive comments on the novelty and importance of this work. We also thank the reviewers for the constructive comments and suggestions that have helped us improve this manuscript.

A point-by-point response to the reviewers' comments is presented below, in blue.

Besides the changes in the manuscript in response to the reviewers' comments, in the course of the revision we realized that three graphs in Supplementary Fig. 3 were mistakenly labelled; the graph labelled *CCP2* showed data for *LCIB*, the graph labelled *CCP1* showed data for *CCP2* and the graph labelled *LCIB* showed data for *CCP1*. We have corrected these errors which had no impact on any of the conclusions of the manuscript.

REVIEWER COMMENTS

Reviewer #1 (Remarks to the Author):

This study used several mutants involved CO₂ metabolism, e.g. the *icl* mutant and *dum11* mutant, to investigate the effects of CO₂ availability on photoprotection in *Chlamydomonas*. The author found that CO₂ availability has a significant role in regulating the expression of *LHCSR1/3* and *PSBS*. Moreover, the CCM regulator *CIA5* controls the expression of *LHCSR3* and *PSBS* whereas inhibits *LHCSR1* protein accumulation. This study demonstrates that CO₂ availability regulates non-photochemical quenching. This is a very exciting finding and extends our understanding of non-photochemical quenching. More importantly, this study shows how light signal and CO₂ signal converge to regulate photoprotection. The experiments were well performed. The conclusions were well supported by the data. In my opinion, the finding in this study is a breakthrough in photoprotection. I have no other comments but suggest the author may emphasize the significance of CO₂ availability controlling photoprotection.

We would like to thank the reviewer for these positive comments on our manuscript and for sharing our excitement in these results.

We believe that the significance of CO₂ availability in controlling photoprotection is sufficiently stressed out throughout the manuscript, e.g.

(i) in the title that reads "Light-independent regulation of algal photoprotection by CO₂ availability",

(ii) in lines 38-40 of the abstract "Here, we show that excess light activates photoprotection- and CCM-related genes by altering intracellular CO₂ concentrations and that depletion of CO₂ drives these responses, even in total darkness",

(iii) in the results section, lines 273-275 "Taken together, our data demonstrate the critical importance of *CIA5* and CO₂ in regulating the different qE effectors, mainly *LHCSR3* and less strongly *PSBS* at the transcript level, and *LHCSR1* at the protein level.",

(iv) in the results section, lines 337-339: "Overall, these data challenge the view concerning the regulation of photoprotection and CCM and bring CO₂ to the forefront as a crucial signal controlling *LHCSR3* and CCM-related genes induction in the absence of light.",

(v) at several instances in the discussion part.

Reviewer #2 (Remarks to the Author):

The manuscript by Ruiz-Sola et al describes a light-independent regulation mechanism of photoprotection by intracellular CO₂ levels. They first show that acetate metabolism could inhibit LHCSR3, an essential component of photoprotection. Moreover, they show that photoprotection genes especially LHCSR3 is regulated by CCM regulator CIA5 and environmental CO₂ levels. This work sheds new light on the crosstalk between CO₂ and light response of *Chlamydomonas* algal cells. Here are some points that may improve the manuscript:

We thank the reviewer for appreciating the novelty of our work on the crosstalk between CO₂ and light responses in the green microalga *Chlamydomonas* and for his/her insightful comments.

Major Points:

1. The biggest concern I have is that authors tried to connect CO₂ availability to photoprotection. However, there is no direct measurement of intracellular CO₂. Also, intracellular CO₂ especially their spatial distribution could be complicated due to the presence of CCM. It might be better to soften some statement, saying that photoprotection proteins are regulated by either acetate metabolism or CCM activities, which may converge to the hypothesis that intracellular CO₂ levels could have a direct impact in photoprotection. In addition, it would be helpful if authors could comment on what technical development might allow people to directly measure spatial-resolved intracellular CO₂ levels, and how it could advance our understanding of photoprotection regulated by CO₂.

We agree with reviewer 2 on this point and we are aware that measuring intracellular CO₂ would be more appropriate for linking intracellular levels of CO₂ and the regulation of NPQ. However, as far as we know, there is no available method for directly measuring intracellular CO₂ concentrations in *Chlamydomonas*.

Having said that, we consider our approach of using intracellular reporter genes as a good proxy to roughly estimate the levels of free CO₂ within the cell, approach also used by others (Hanawa et al., 2007). Our results show that: (i) the *icl* mutant, which is unable to metabolize acetate (Fig. 2c), exhibits *LHCSR3* downregulation in the presence of CO₂ (Fig. 2a), demonstrating that CO₂ can substitute for acetate metabolism in suppressing *LHCSR3* expression; (ii) this repression by acetate is much stronger in low light (Fig. 2a), where CO₂ fixation is slow, than in high light (Fig. 2d), where CO₂ fixation is much faster, supporting the idea that acetate represses *LHCSR3* via acetate-derived CO₂; (iii) the DCMU-dependent reduction of *LHCSR3* mRNA accumulation can be reversed by sparging the cells with very low CO₂ (VLCO₂; Fig. 6b), indicating that intracellular CO₂ accumulation (indirectly measured as an increase in CO₂ released by cells; Fig. 6a) is mediating *LHCSR3* downregulation; (iv) *LHCSR3* mRNA accumulation is induced after reducing CO₂ in the medium even in the absence of light (Fig. 5a), demonstrating that NPQ can respond directly to the CO₂ levels; light is not directly required for modulating the level of the *LHCSR3* transcript; (v) Our model further supports the findings that acetate metabolism would increase the intracellular CO₂ concentration (Supplementary Note, Supplementary Fig. 2) to a level that could suppress expression of *LHCSR3*. Interestingly, application of nanoscale secondary ion mass spectrometry (NanoSIMS) technology in *Chlamydomonas* cells incubated with ¹³C-labelled

acetate revealed an enrichment of ^{13}C signal in the pyrenoid, likely reflecting fixation of acetate-derived CO_2 (Penen et al. 2020).

Together, all of these results strongly indicate that *LHCSR3* expression is impacted by CO_2 itself, even in the complete absence of light, and that the altered *LHCSR3* level in turn, would regulate NPQ through a mechanism that involves CIA5 (Fig. 3 and 5).

Whether different CCM activities (carbonic anhydrases, bicarbonate transporters, etc.) have a direct role in NPQ regulation is an interesting but difficult question to answer, as dissecting CO_2 levels and CCM activities would probably entail the study of NPQ regulation in individual CCM mutants, something that would require an enormous amount of time and resources and still not be conclusive because the phenotypes of some of the mutants might be the result of secondary effects elicited by the lesions. Based on the current evidence, the results are in accord with the CCM enabling the cells to concentrate CO_2 , which in turn would generally suppress quenching (the level of *LHCSR3* would decline).

Time-resolved measurements of spatial-resolved intracellular CO_2 levels would further advance our understanding of how changing light and CO_2 impacts microalgae physiology. In this direction, NanoSIMS technology was used to trace subcellular localization of ^{13}C -bicarbonate within a photosymbiotic sponge (Achlati et al., 2018) while bicarbonate levels were probed using a soluble cyclase/cAMP-based fluorescent FRET biosensor in animal cells (Bernhard et al. 2020). However, as NanoSIMS is a disruptive approach unable to capture dynamic changes in bicarbonate levels, and the expression of the biosensor caused severe cell stress and subsequent cell death, the need for technological breakthroughs to address the spatial measurements of inorganic carbon in *Chlamydomonas* remains.

Finally, to reach a full understanding of the regulation of photoprotection by CO_2 , signalling components linking CO_2 sensing with transcriptional regulation of photoprotection genes need to be identified. We discuss about that in lines 435-448; we believe that future research will shed more light into this open question.

References

Achlati, M., Pernice, M., Green, K., Guagliardo, P., Kilburn, M.R., Hoegh-Guldberg, O. & Dove, S. Single-cell measurement of ammonium and bicarbonate uptake within a photosymbiotic bioeroding sponge. *The ISME Journal* **12**, 1308–1318 (2018).

Bernhard, K., Stahl, C., Martens, R., & Frey, M. A Novel Genetically Encoded Single Use Sensory Cellular Test System Measures Bicarbonate Concentration Changes in Living Cells. *Sensors* **20**, 20, 1570 (2020)

Hanawa, Y., Watanabe, M., Karatsu, Y., Fukuzawa, H. & Shiraiwa, Y. Induction of a high- CO_2 -inducible, periplasmic protein, H43, and its application as a high- CO_2 -responsive marker for study of the high- CO_2 -sensing mechanism in *Chlamydomonas reinhardtii*. *Plant Cell Physiol.* **48**, 299–309 (2007).

Penen, F. Isaure, L.-P., Dobritsch, D., Castillo-Michel, H., Gontier, E., Le Coustumer, P., Malherbe, J. & Schaumlöffel, D. Pyrenoidal sequestration of cadmium impairs carbon dioxide fixation in a microalga. *Plant Cell Physiol.* **43**, 479–495 (2020).

2. Why in Fig. 1a, in icl complement in LL, acetate shows similar level of LHCSR3 mRNA as in air? Does the complementation strain has lower level of ICL compared to wildtype? Or it's LL so that difference might be small?

We thank the reviewer for this observation. To better understand why *icl-C* strain behaved differently than the WT in LL conditions, we quantified *ICL* gene expression in WT, *icl* and *icl-C* in LL from the cDNA samples that we used to analyze *LHCSR3* mRNA levels in Fig. 1a. The results (see below, Rebuttal Fig. 1) show that WT and *icl-C* strains accumulate similar levels of *ICL* mRNA, in line with the original publication describing the generation of the *icl* and *icl-C* strains (Plancke, C. *et al. Plant J.* 77, 404–417 (2014)).

Why does *icl-C* not behave exactly like WT in LL? We do not have a satisfying answer to this question. It is conceivable that the similar levels of *LHCSR3* mRNA (air vs acetate) that we observe in LL acclimated *icl-C* in the presence or absence of acetate are due to the fact that the *LHCSR3* mRNA levels are very low under low light conditions for all of the strains under all conditions tested. Since the absolute values are very low, relatively small changes in measured values can result in variability of *LHCSR3* mRNA levels and the ratios of the mRNA between conditions; this could also be impacted by small differences in the growth/physiological state of the different strains that are being used. So when measuring very low mRNA levels in different strains, changes in measured absolute values can make it more difficult to make precise comparisons.

Rebuttal Fig. 1: mRNA accumulation of *ICL* in WT and *icl-C* strains was analyzed in the very same cDNA samples of Fig. 1a. The mRNA levels of *ICL* were undetectable in the *icl* mutant and therefore are not shown in the graph. The p-values for the comparisons of WT and *icl-C* under the different conditions (air, acetate, CO₂) are based on ANOVA Dunnett's multiple comparisons test of log10-transformed mRNA data as indicated in the graphs (*, P < 0.005; ns, not significant).

3. In some cases, the complement strains do not always recapitulate the phenotype of wildtype. For example *icl-C* in Fig. 1a in LL and *cia5-C* in fig 3c. Have you considered comparing the protein or mRNA level of the protein in WT and complement strains? Or try using the native promoter? Or do you have other assays like spot test to show the complementation actually works?

Concerning the *icl-C* in Fig. 1a, please see our response to the abovementioned point 2.

As far as *cia5-C* is concerned, complementation of the *cia5* with the *CIA5* gene successfully rescued the gene expression, protein accumulation and qE phenotypes, described in the manuscript (e.g. in Figs. 3 and 5), so we are confident of the success of the complementation. We have also

addressed how the pre-acclimation conditions (dark vs. low light) impacted CIA5 protein accumulation and the recovery of the LHCSR1 phenotype (Lines 323-336 in the manuscript).

In the case of Fig. 3c, *cia5-C* indeed developed higher qE than the WT; this could indicate higher CIA5 protein levels in the *cia5-C* strain compared to the WT, perhaps because of the use of the strong *PSAD* promoter instead of the endogenous *CIA5* promoter; yet this is not clearly supported by the protein levels of LHCSR3 and LHCSR1 (Fig. 3b). We agree with the reviewer that a spot test would add further confidence that the complementation actually works. We performed spot tests on HSM agar under medium light intensity ($100 \mu\text{mol photons m}^{-2} \text{s}^{-1}$) and could confirm what was expected, i.e. that the *cia5* mutant is an air-dyer. In the same test we could also further validate that this phenotype is rescued by ectopic expression of *CIA5* in the *cia5-C* strain. This new dataset is presented in Supplementary Fig. 7c and in lines 500-501 of the Materials and Methods section.

4. What is the physiological significance that other than light, CO₂ could also regulate photoprotection? Does it mean that the optimal light intensity will be set by CO₂ fixation capacity. Once above the optimal intensity, excessive light will be dissipated as heat.

The concept of optimal light intensity is hard to define due to the fluctuating nature of light in natural environments. Assuming that the optimal light intensity is defined as the light intensity upon which the organism grows at its maximal speed, then yes, the optimal light intensity will be set by CO₂ fixation capacity. CO₂ regulation of NPQ components highlights a possible antagonistic interaction between photosynthesis and photoprotection; high CO₂ availability would repress photoprotection (NPQ), favoring photochemical quenching over NPQ; low-CO₂ availability would stimulate photoprotection, which would avoid oversaturation of the electron transfer chain and production of reactive oxygen species. Accordingly, we discuss the following in lines 450-455: “Exposure to HL increases the CO₂ fixation rate which causes a drop in the intracellular CO₂ level, which in turn, activates both photoprotection- and CCM-related genes. This depletion of CO₂ is sufficient to drive high levels of expression of the CCM and *LHCSR3* genes even in complete darkness. On the other hand, high CO₂ levels, either generated through enhanced respiratory activity or impaired photosynthetic electron transport, repress *LHCSR3* and CCM genes while at the same time stabilizing the LHCSR1 protein”.

5. Could the authors comment more on the relationship between LHCSR1 and LHCSR3, in the context of different CO₂ and light conditions. It seems to me that LHCSR3 is predominant in photoprotection function, longer lasting after light exposure, but much more sensitive to different CO₂ levels. On the contrary, LHCSR1 is more independent with CO₂ levels, and could serve as a backup photoprotection protein.

The reviewer raises a very important point; LHCSR1 does seem to have a compensatory role, providing the cells with photoprotection under conditions where LHCSR3, the key photoprotective protein is absent. This was mentioned in the original submission in the “Results” section, Lines 244-250 and in the “Discussion” section lines 453-456. We now slightly extended the discussion by adding the grey highlighted part of the text (see below).

Lines 244-250: “Our data additionally suggest that accumulation of LHCSR1 protein occurs through a compensatory, CIA5-controlled posttranscriptional mechanism that provides

photoprotection under conditions in which the cells have almost no LHCSR3 protein (compare LHCSR1 and LHCSR3 immunoblots in Fig. 3b). Supporting this idea, the qE levels in *cia5*, although lower than WT and *cia5-C* (Fig. 3c and Supplementary Fig. 4), were unexpectedly high considering the absence of LHCSR3 protein (Fig. 3b); we attribute this result to overaccumulation of LHCSR1 in this mutant (Fig. 3b).”

Lines 453-456: On the other hand, high CO₂ levels, either generated through enhanced respiratory activity or impaired photosynthetic electron transport, repress LHCSR3 and CCM genes while at the same time stabilizing the LHCSR1 protein, which likely acts as a backup photoprotection protein under conditions under which LHCSR3 is not expressed.

Minor points:

1. In the section “Link between photosynthetic electron transfer and CO₂ intracellular concentration”, the full name of PET should be written at least once.

Thank you for pointing this out, we have spelled out PET at its first instance.

2. In Fig. 2a, it is hard to distinguish air and CO₂ based on color. Consider making the line thicker in the legend, or use colors that are more distinguishable. Also in the legend of Fig 2a, the font size of “air” is bigger than “acet”

Fig. 2 has been amended to improve readability.

Reviewer #3 (Remarks to the Author):

I have read the manuscript "Light-independent regulation of algal photoprotection by CO₂ availability" by Ruiz-Sola et al. in which the authors analysed the molecular mechanisms regulating the process of CO₂ concentrating mechanisms (CCM) and photoprotection in *Chlamydomonas reinhardtii*. To this aim, the authors have used genetic and mathematical modelling approaches and show (among other things) that inhibition of LHCSR3 accumulation and CCM activity by acetate is at the level of transcription and a consequence of metabolically produced CO₂. My expertise in computational modelling, so my comments mostly refer to that part of the work. The authors have used constraint-based metabolic modelling to assess whether there are changes in the internal concentration of CO₂ under different trophic conditions and at different light intensities. The analyses performed are standard and take advantage of already available functions from the cobra toolbox. Overall the optimization problems are clearly defined and seem to have a well-grounded biological counterpart. The analysis could have provided more insights into this problem, particularly going at the single reaction-level.

We thank the reviewer for acknowledging that the procedure is well described and integrates knowledge from the other biological experiments in this study. We have added a more detailed analyses of the findings at the level of individual pathways/ reactions as detailed below.

First, they used FVA to extract those reactions whose steady state flux ranges did not overlap

between the WT and modelled mutants. However, it is not clear how this set of reactions was identified. Were these reactions not overlapping at all in their allowed flux ranges to maintain the solution optimal or some tolerance was introduced? The authors provide a supplementary table with all the flux distributions obtained in their simulations but it is not straightforward to derive the criteria adopted in each circumstance.

Flux ranges were considered non-overlapping if:

- (1) the minimum flux obtained from FVA in the WT is above the maximum flux obtained for mutants or if the minimum flux obtained from FVA in the mutant is above the maximum flux obtained from FVA i.e. there is no intersection between the flux ranges;
- (2) minimum flux for both, WT and mutants, is greater than $0.01 \text{ mmol gDW}^{-1} \text{ h}^{-1}$, to avoid considering reactions of low absolute flux.
- (3) in line with differential expression analysis, where one considers genes differentially expressed above a preselected fold-change (e.g. of at least 2, in nominal values), for the flux ranges that do not overlap, we use a threshold on the relative difference between the lower bounds of at least 5% (we used 5% to be less restrictive) to filter for cases where flux ranges are close to each other; this condition is meant to remove any numerical artifacts.

We included a description of the procedure in the Supplementary Note.

They also assessed that set of reactions showing differences in all the simulated conditions are enriched with CO₂-producing reactions. It is not clear, however, whether the set of CO₂ producing reactions remain the same and their fluxes increase or, rather, other CO₂ producing reactions come into play (providing additional intracellular CO₂). Authors could have used the computeFluxSplits to compute relative contributions of fluxes to the net production and consumption of CO₂ and check the CO₂ metabolism at the single reaction level. Are there specific reactions that contribute more than the others to the CO₂ intracellular increase?

The underlying set of active (i.e. of non-zero flux) CO₂ producing reactions is the same for WT, *icl* and *dum11*. We add a sentence to the related section that clarifies that Fisher's exact test is based on the same set of active CO₂-producing reactions.

In addition, we followed the reviewer's suggestion and computed relative contribution of flux to production of CO₂ (Supplementary Data 1d). We observe no difference in the relative contribution that could explain the phenotypic differences across the three conditions.

Also, it would be interesting to know how downstream pathways react to this increase in CO₂ availability as, for example, Calvin-Benson-Bassham cycle. Again, the information is already there but it appears under-investigated.

Following the suggestion of the reviewer, we compared differential behavior on a pathway level. To this end, we use the information on model subsystems provided along the model reconstruction and investigate the percentage of reactions per model pathway that show significant change in sampled flux values in both mutants, *icl* and *dum11* in comparison to the WT for the respective conditions. The results are described in the Supplementary Note and shown in Supplementary Data 1c.

I am a bit surprised by the choice of following the expression of 12 selected CCM genes rather than performing a whole-cell transcriptomic analysis. I understand that the target of this work

resides in the analysis of this specific cellular process but an -omic approach here would have allowed to provide a more general view of the problem, besides answering the specific question posed here by the author. Also, it would have probably been able to suggest other putative CIA5 targets and, as such, pave the way to further analyses.

We can only agree with the reviewer's suggestion; applying an -omic approach would allow us to uncover the wider view of the role of light and CO₂ in the regulation of gene expression. This was stressed in the concluding sentences of the discussion (lines 459-462) and this is what we are currently pursuing in an on-going project in the lab. However, while an omics approach is important, we consider it beyond the scope of the present study.

Reviewer #2 (Remarks to the Author):

The response from the authors is satisfactory. All the points I raised were properly addressed and answered. Overall this manuscript is eye-opening, solid and well written.

Reviewer #3 (Remarks to the Author):

The authors have carefully addressed my concerns. Given the outcomes of these analyses I was expecting some more discussion about them, specifically when they addressed the following point:

"Also, it would be interesting to know how downstream pathways react to this increase in CO₂ availability as, for example, Calvin-Benson-Bassham cycle. Again, the information is already there but it appears under-investigated."

They indeed found a few pathways that were apparently influenced by CO₂ availability (N-glycan, protein biosynthesis, etc.) but did not include any explanation on the biological reason of such reprogramming. It would be much more interesting to include some mechanistic explanation of such flux variations.

Response to the reviewers' comments

Reviewer #2 (Remarks to the Author):

The response from the authors is satisfactory. All the points I raised were properly addressed and answered. Overall this manuscript is eye-opening, solid and well written.

- We thank the reviewer for these very supportive comments on our work.

Reviewer #3 (Remarks to the Author):

The authors have carefully addressed my concerns. Given the outcomes of these analyses I was expecting some more discussion about them, specifically when they addressed the following point:

"Also, it would be interesting to know how downstream pathways react to this increase in CO₂ availability as, for example, Calvin-Benson-Bassham cycle. Again, the information is already there but it appears under-investigated."

They indeed found a few pathways that were apparently influenced by CO₂ availability (N-glycan, protein biosynthesis, etc.) but did not include any explanation on the biological reason of such reprogramming. It would be much more interesting to include some mechanistic explanation of such flux variations.

- We are pleased to read that the revised manuscript addressed the reviewer's previous concerns and we thank the reviewer for raising this interesting point. We have now added the following text to address this point:

"These observed changes in fluxes may be explained by transcriptional reprogramming that affect downstream enzyme abundances who support the flux changes. As demonstrated in our experimental validation, CO₂ can serve as a signal for these transcriptional reprogramming. In addition, other mechanisms related to allosteric regulation of reaction rates cannot be excluded."